# Merkel Cell Carcinoma Display PIEZO2 Immunoreactivity

**DOI:** 10.3390/jpm12060894

**Published:** 2022-05-28

**Authors:** Yolanda García-Mesa, Raquel Martín-Sanz, Jorge García-Piqueras, Ramón Cobo, Saray Muñoz-Bravo, Olivia García-Suárez, Benjamín Martín-Biedma, José Antonio Vega, Jorge Feito

**Affiliations:** 1Departamento de Morfología y Biología Celular, Grupo SINPOS, Universidad de Oviedo, 33003 Oviedo, Spain; garciamyolanda@uniovi.es (Y.G.-M.); jgarciap@unizar.es (J.G.-P.); ramoncobodiaz@gmail.com (R.C.); garciaolivia@uniovi.es (O.G.-S.); javega@uniovi.es (J.A.V.); 2Servicio de Oftalmología, IBSAL, Complejo Asistencial Universitario de Salamanca, 37007 Salamanca, Spain; rmartinsan@saludcastillayleon.es; 3Departamento de Anatomía e Histología, Universidad de Zaragoza, 50009 Zaragoza, Spain; 4Servicio de Anatomía Patológica, Instituto de Investigación Biomédica de Salamanca, Complejo Asistencial Universitario de Salamanca, 37007 Salamanca, Spain; smunozb@saludcastillayleon.es; 5Departamento de Cirugía y Especialidades Médico-Quirúrgicas, Universidad de Santiago de Compostela, 15782 Santiago de Compostela, Spain; benjamin.martin@usc.es; 6Facultad de Ciencias de la Salud, Universidad Autónoma de Chile, 7500912 Santiago, Chile

**Keywords:** merkel cells, merkel cell carcinoma, PIEZO2, mechanobiology, cancer mechanobiology, ion channels

## Abstract

As an essential component of mechano-gated ion channels, critically required for mechanotransduction in mammalian cells, PIEZO2 is known to be characteristically expressed by Merkel cells in human skin. Here, we immunohistochemically investigated the occurrence of Piezo channels in a case series of Merkel cell carcinoma. A panel of antibodies was used to characterize Merkel cells, and to detect PIEZO2 expression. All analyzed tumors displayed PIEZO2 in nearly all cells, showing two patterns of immunostaining: membranous and perinuclear dot-like. PIEZO2 co-localized with cytokeratin 20, chromogranin A, synaptophysin and neurofilament. Moreover, neurofilament immunoreactive structures resembling nerve-Merkel cell contacts were occasionally found. PIEZO2 was also detected in cells of the sweat ducts. The role of PIEZO2 in Merkel cell carcinoma is still unknown, but it could be related with the mechanical regulation of the tumor biology or be a mere vestige of the Merkel cell derivation.

## 1. Introduction

Merkel cell carcinoma (MCC) is a rare, clinically aggressive neuroendocrine tumor of the skin with high propensity for local, regional, and distant spread [1,2,3,4], being lethal in about 3–35% of cases [5]. MCC fundamentally qoccurs in elderly people, being sun exposure or immunosuppression known risk factors [6]. MCC nature has still some controversy: Sunshine et al. [7] recently hypothesized both Merkel cells (MC) and dermal fibroblasts might be on the cellular origin. Epidermal stem cells, keratinocytes and Pro-B or Pre-B cells have been also proposed as the cellular origin [2]. In any case, most evidences lend support to an origin from MC [8]. The etiology for MCC is currently divided between Merkel cell polyomavirus (MCPyV) and solar exposure with UV-induced mutations [9,10].

MCs are highly specialized epithelial cells located in the epidermal basal layer and also in the external portion of hair follicles. Classically, they were regarded as nonkeratinocyte epidermal “tastzellen” or “touch cell” that functions as a tactile skin receptor [11]. Currently, it has been definitively demonstrated that they are essential components of the Merkel cell-neurite complexes acting together as type I slowly-adapting low-threshold mechanoreceptors by transforming tactile stimuli into Ca2^+^-action potentials [12], inducing release of neurotransmitters and activating Aβ-afferent nerve endings [13,14].

Mechanically gated ion channels are at the origin of these mechanically induced action potentials, in a process called mechanotransduction [15]. Among them, members of PIEZO family are essential components of distinct stretch-activated ion channels capable to perform mechanotransduction thanks to their particular tridimensional arrangement [16], being expressed in a wide range of normal and neoplastic tissues [17,18,19]. PIEZO proteins represent a new class of mechanosensitive channels, which respond to mechanical forces and allow Ca2^+^ influx in the cell. PIEZO1 and PIEZO2 are the only members of PIEZO family, characterized by a high grade of homology and a similar mechanosensory function [18,19]. They are assumed (especially PIEZO1) to participate in various mechano-associated biologic processes such as sensing of shear stress (particularly in the vasculature), bladder distention and regulation of urine flow, volume regulation, cellular development, proliferation, migration and elongation [19,20]. 

Although both PIEZO channels share a great homology, PIEZO1 is a more polymodal sensor for mechanical forces, detecting a larger number of stimuli, while PIEZO2 is more narrowly tuned to specifically detect mechanical touch [19]. Thus, PIEZO2 is present in peripheral sensory neurons and cutaneous low-threshold mechanoreceptors [21,22,23], as well as in MCs [13,22,24,25]. Particularly regarding MCs, both cellular density and PIEZO2 expression diminish with age in glabrous digital skin [26], which makes contrast with the elderly orientation of MCC. Presumably the behavior of MC is different in other anatomical placements. Whether MCC also express PIEZO proteins has been never investigated, although PIEZO2 presence in cultured human MCC-13 cells has been probed [27]. Therefore, the present study was designed to investigate the occurrence of PIEZO2 in MCC. This could be of potential interest because of the potential role of mechanosensitive ion channels in diagnosis, prognosis or treatment of cancer [15,16,28].

## 2. Materials and Methods

Samples of histologically diagnosed MCC were analyzed to perform the research. The localization of the tumors was predominantly the head (n = 7, corresponding to ear, parietal region, eyelid, nose and cheek), arm (forearm and hand, n = 3) and leg (n = 3). The age range was between 70 and 92 years (8 females and 5 males). All of these materials were obtained in compliance with The Spanish Law (RD 1301/2006; Ley 14/2007; DR 1716/2011; Orden ECC 1414/2013) and the study was approved by the Ethical Committee for Biomedical Research of the Complejo Asistencial Universitario de Salamanca, Spain (Cod. CElm: PI2022 02 935). The specimens were fixed in 4% formaldehyde (0.1 M phosphate-buffered saline; pH 7.4) and embedded in paraffin as usual. The pieces were cut perpendicularly to the skin surface into 10 µm thick sections and mounted on gelatin-coated microscope slides. 

Immunohistochemistry was performed using the automated diagnostic platform Leica Bond III with the Leica Bond™ Polymer Refine Detection Kit (Leica Biosystems™, Newcastle upon Tyne, UK), following the manufacturer’s instructions.

Primary antibodies used in the study are listed in Table 1. Antibodies against cytokeratin 20 (CK20), synaptophysin (Syn), chromogranin A (ChrA), and neurofilament proteins (NFP) were used to identify and characterize MC. Sections were incubated with primary antibodies for 1 overnight at 4 °C in a dark humid chamber. After rinsing, Dako EnVision System labeled polymer-HR anti-rabbit IgG or anti-mouse IgG (DakoCytomation, Glostrup, Denmark) was applied for 30 min at room temperature. Immunoreaction was visualized by using 3-3′-diaminobenzidine as chromogen. Finally, sections were counterstained with hematoxylin-eosin in order to ascertain structural details. For control purpose, sections were incubated without primary antibodies and/or secondary antibodies, employing non-immune rabbit/mouse sera instead primary antibodies.

On the other hand, double immunofluorescence was performed in order to investigate the co-localization of PIEZO2 with the above MC markers, as well as the possible relationship of MCC cells with neural structures. Skin sections were deparaffinized and rehydrated, and the non-specific binding was reduced by bovine serum albumin (5% in Tris Buffer Saline; pH 7,4) for 30 min. Sections were incubated overnight, at 4 °C, in a dark humid chamber with a 1:1 *v*/*v* mixture of anti-PIEZO2 and anti-CK20; or anti-PIEZO2 and anti-Syn; or anti-PIEZO2 and anti-ChrA; and anti-PIEZO2 and NFP. Once the sections were rinsed, they were incubated with Alexa fluor 488-conjugated goat anti-rabbit IgG (Serotec™, Oxford, UK, diluted 1:1000) and Cy 3-conjugated donkey anti-mouse antibody (Jackson-ImmunoResearch™, Baltimore, MD, USA, diluted 1:50), one after another, in a dark humid chamber, for 1 h, at room temperature. Finally, nuclei were labelled with DAPI (10 ng/mL). Immunofluorescence was detected by using a Leica DMR-XA automatic fluorescence microscope coupled with a Leica Confocal Software, version 2.5 (Leica Microsystems, Heidelberg GmbH, Heidelberg, Germany). Additionally, control sections were processed as previously described, but primary antibodies were either omitted in the incubation or substituted by non-immune rabbit/mouse sera.

## 3. Results

Merkel cells from human skin displayed immunoreactivity for CK20 and ChrA, which well co-localized with PIEZO2 (Appendix A). Structurally MCC formed cords, trabeculae or sheets of small monomorphic cells that occasionally contained blood vessels. The assessment and diagnosis of MCC was performed based on the expression of markers such as CK20 (Figure 1a,b), chromogranin A (Figure 1c,d), synaptophysin (Figure 1e,f) or NFP (Figure 1g,h), although the proportion of NFP positive cells was generally low. 

PIEZO2 immunoreactivity was detected in all the cases, with a characteristic cytoplasmic pattern of distribution in the cells expressing CK20 (Figure 2a,b). The general pattern consisted in an overlap between membranous (Figure 2c–f and Figure 3a,b) and paranuclear dot-like (Figure 2g,h and Figure 3c,d) patterns. 

PIEZO2 immunoreactivity was also identified employing immunonofluorescence (Figure 3). Using double immunohistochemistry for PIEZO2 and CK20 (Figure 3e), PIEZO2 and ChrA (Figure 3f), or PIEZO2 and NFP (Figure 3g–j), we observed co-localization of those proteins in a variable proportion of cells within the tumor mass. 

In addition to MC of the basal epidermal layer and the neoplastic cell mass of MCC, PIEZO2 immunoreactivity was detected in some other cutaneous structures, primarily in ducts of sweat and sebaceous glands. It was observed in nearly all ductal superficial cells and not in basal epithelial elements, morphologically identified as myoepithelial cells (Figure 4a,d). Furthermore, these PIEZO2-positive cells, in contrast with most cells in MCC, were in contact with nerve profiles (Figure 4b,c,e,f).

## 4. Discussion

Although it has been definitively established that MCs are specialized epithelial cells [29,30], they display a mixed immunohistochemical profile proper of epithelial and neuroendocrine cells [2]. The common ectodermal origin of both neuroendocrine and epidermal lineages might explain that mixed phenotype. Here, we demonstrate that, similarly to MCs, MCC display PIEZO2 immunoreactivity, with a similar dot-like pattern presented by other antigens such as intermediate filaments, typical of neuroendocrine malignancies [10,31]. The expression of this MCs essential protein is another of the many features shared by MCs and MCC cells. To the best of our understanding, the occurrence of PIEZO proteins in MCC was never demonstrated, although it was reported in human MCC-13 cell line [27]. 

In neoplastic tissue, the role of PIEZO channels, in particular PIEZO2, is still under study [15]. Furthermore, both extracellular matrix and intracellular cytoskeleton are involved in PIEZO2 function in addition to neoplastic cells [16]. The stroma is implicated indirectly, by maintaining the electrochemical gradient necessary for Ca2^+^ influx (for example, K2P, KCa channels) [28]. In addition, the precise composition of the cellular lipid bilayer may be relevant for mechanotransduction [16]. Mechanical forces stimulate these channels both pushing and pulling the cell membrane through the extracellular matrix, whose presence is required for a correct function of PIEZO channels [32]. These forces may influence cancer biology by affecting both neoplastic cells and their microenvironment, by altering cell migration, proliferation, extracellular matrix remodelling and metastatic spread [28]. Moreover, integrin activation on the cell surface and adhesion to other cells and extracellular matrix depends on the presence of PIEZO1 channels [33]. However, although PIEZO channels have emerged as the key element in mechanotransduction, several types of mechanosensitive ion channels are also involved in this ability, and even other channels may be decisive for mechanotransduction in addition to PIEZO [34]. 

MCC is known for its poor prognosis and aggressive behavior and exhibits not only PIEZO2 expression in the MCC-13 cell line [27], but also widespread expression on all the tumors studied in our work. PIEZO2 could be significant in the progression of MCC, but if this has clinical relevance or is a mere remnant of the Merkel cell where the MCC developed, remains to be demonstrated in future studies using larger series searching for PIEZO2-negative MCC cases, to determine if this protein define a subgroup of MCC with different clinical or molecular characteristics. In any case, although in the presented MCC cases there is no evidence of a prognostic value of PIEZO2, MCC could benefit from a possible PIEZO related therapy, as long as both types of PIEZO channels are known to be inhibited by the toxin GsMTx4 and stimulated by Yoda1 [34]. Other promising therapies are in study or already in use, such as PD-1/PDL-1 pathway blocking agents, multi-targeted tyrosine kinase inhibitors, MCPyV-related immunotherapy and AURKA inhibitors, potentially useful in MCPyV-negative MCC, among others [9,10,35,36].

## Figures and Tables

**Figure 1 jpm-12-00894-f001:**
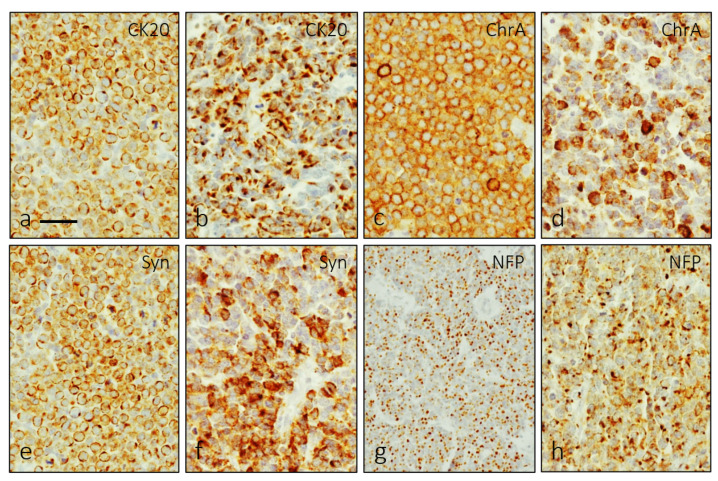
Single immunohistochemistry for CK20 (**a**,**b**), ChrA (**c**,**d**), Syn (**e**,**f**) and NFP (**g**,**h**) was used to characterize MCC cells. They showed two patterns of immunostaining: membranous perinuclear and dot-like. Scale bar 20 µm.

**Figure 2 jpm-12-00894-f002:**
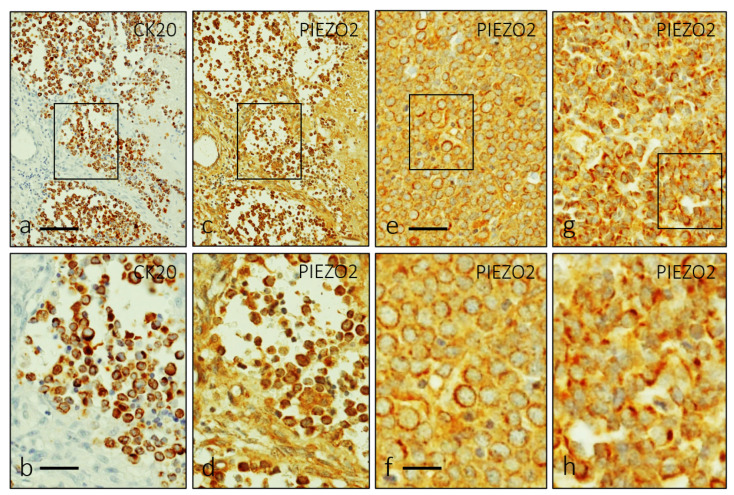
Single immunohistochemistry for CK20 (**a**,**b**) and PIEZO2 (**c**–**h**) in approximate serial sections showing a perinuclear pattern of distribution identical for both proteins. A detailed examination reveals two patterns of PIEZO2 immunostaining: cytoplasmic with perinuclear halo (**e**,**f**) and dot-like (**g**,**h**). Scale bars 100 µm (**a**,**c**,**e**,**g**), 50 µm (**b**,**d**), 20 µm (**f**,**h**).

**Figure 3 jpm-12-00894-f003:**
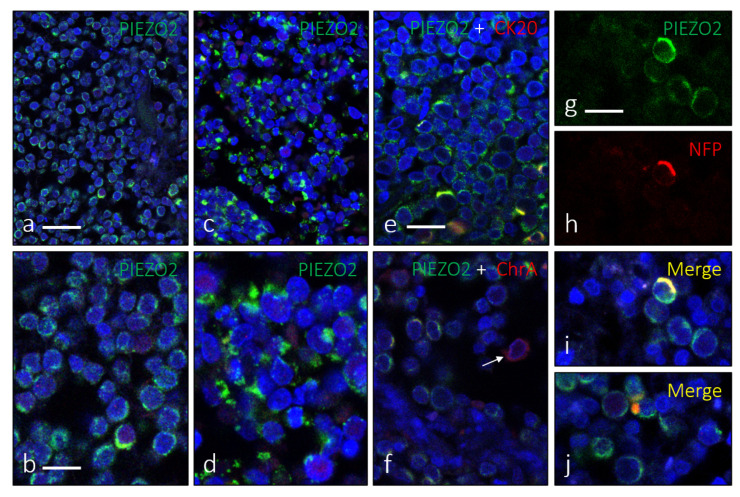
Immunofluorescence for PIEZO2, showing its two morphological patterns: cytoplasmic perinuclear (**a**,**b**) and dot-like pattern (**c**,**d**). Co-localization of PIEZO2 with CK20 (**e**) or ChrA (**f**) was regularly observed (merge in yellow). Arrow in ‘f’ indicates a ChrA+/PIEZO2− cell. The MCC cells were processed for simultaneous detection of PIEZO2 and NFP (**g**–**j**) only revealed a scarce number of cells showing co-localization of both (**i**), sometimes resembling MC-nerve contact (arrow in **j**). Scale bars 50 µm (**a**,**c**), 20 µm (**b**,**d**–**j**).

**Figure 4 jpm-12-00894-f004:**
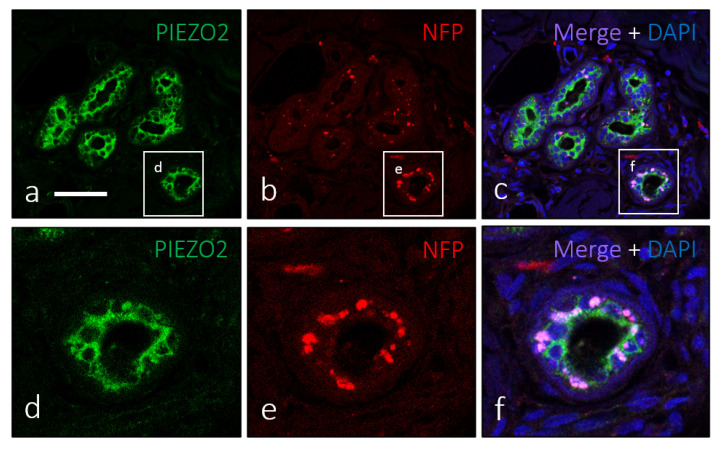
The ductal cells of sweat glands show PIEZO2 immunostaining (**a**,**d**) and were densely innervated (**b**,**c**,**e**,**f**). Scale bar 100 µm.

**Table 1 jpm-12-00894-t001:** Primary antibodies used in the study.

Antigen	Origin	Dilution	Supplier
Merkel cell markers			
CK20 (clone Ks20.8-, IS777)	Mouse	Prediluted	Dako ^1^
Chromogranin A	Mouse	Prediluted	Leica ^2^
NFP (clone 2F11)	Mouse	1:100	Dako ^1^
Synaptophysin (clone DAK-SYNAP)	Mouse	1:100	Dako ^1^
PIEZO2	Rabbit	1:200	Milipore Sigma ^3^

CK20: cytokeratin 20; NFP: neurofilament. ^1^ Glostrup, Denmark; ^2^ Newcastle upon Tyne, United Kingdom; ^3^ Burlington, MA, USA.

## Data Availability

Not applicable.

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
