# Peer review of "Merkel Cell Carcinoma Display PIEZO2 Immunoreactivity"

_jpm, 2022, doi:10.3390/jpm12060894_

Round 1
Reviewer 1 Report
In the introduction you state that MCs, both cellular density and Piezo2 expression, diminish with age, which makes contrast with the elderly orientation of MCC. Could you elaborate what you mean, particularly regarding the correlation between Piezo2 expression and the old age at diagnosis in MCC?
Could you clarify the interpretation method and threshold for Piezo 1 and Piezo2 immunohistochemistry? Could you further clarify the extent of staining of both antibodies in Merkel cell carcinomas?
While in the results you state that Piezo1 immunohistochemistry overlaps with Piezo2, both in the materials and methods and in the discussion you elaborate only on Piezo2. Could you clarify this point?
You don’t mention MCPyV status in the examined series of MCC. Have you tried, or do you plan to try to correlate Piezo2 immunohistochemistry with MCPyV large T antigen immunohistochemistry in MCC?
Could you elaborate on the potential Piezo related therapies?
Author Response
Thank you for your interest and support in the work, your suggestions are highly appreciated, and we would like to respond to all of them.
“In the introduction you state that MCs, both cellular density and Piezo2 expression, diminish with age, which makes contrast with the elderly orientation of MCC. Could you elaborate what you mean, particularly regarding the correlation between Piezo2 expression and the old age at diagnosis in MCC?”
- Age dependent relationships were not described. The described fact is, indeed, that Piezo2 positive Merkel cells diminish with age, which doesn´t necessarily mean a diminution in Piezo2 itself. In any case we clarified this point in the text.
“Could you clarify the interpretation method and threshold for Piezo 1 and Piezo2 immunohistochemistry? Could you further clarify the extent of staining of both antibodies in Merkel cell carcinomas?”
“While in the results you state that Piezo1 immunohistochemistry overlaps with Piezo2, both in the materials and methods and in the discussion you elaborate only on Piezo2. Could you clarify this point?”
- We removed the references to Piezo1, as long as it´s expression was not unequivocally proved in Merkel cells.
“You don’t mention MCPyV status in the examined series of MCC. Have you tried, or do you plan to try to correlate Piezo2 immunohistochemistry with MCPyV large T antigen immunohistochemistry in MCC?”
“Could you elaborate on the potential Piezo related therapies? “
- In the initial design we expected some degree of variability in the Piezo2 expression. This would have given us opportunity of exploring the polyomavirus status, but also a potential clinical variance. As long as the finding were quite uniform, we think these correlations have little significance and we considered unnecesary. Directly correlating MCPyV large T antigen with Piezo is a quite interesting idea, honestly we are currently exploring a convincing Piezo1 demonstration, and we will probably include the proposed correlation. We extended the last paragraph of discussion, to mention the existence of Piezo inhibitors and stimulators, also indicating the most important recent MCC therapies. Moreover, in the introduction we briefly mention now the MCPyV significance as a main causal agent.
Reviewer 2 Report
The authors describe the expression of Piezo channels in Merkel cell carcinoma. These markers were identified in Merkel cells, but this is the first study to address the question of whether MCC cells express these channels. This is an interesting study since novel biomarkers for diseases are potential targets for new drugs.
Page 1, line 38: what is “pre/pro”? Would it be pre-B cells and pro-B cells?
Page 2, lines 67-79: “This could be of potential interest because the role of mechanosensitive ion channels in cancer, with diagnostic or prognostic purposes, but also a noticeable potential in therapy”. This sentence should be reformulated: “This could be of potential interest because of the role of mechanosensitive ion channels in diagnostic, prognostic, treatment in cancer”
Figure 2. The legend says that CK20 is demonstrated in “a” and “b”, but images show “b” as Piezo 2, and “e” as CK20. The same error is on the text (line 146)
Figure 3. The legend says that CK20 is demonstrated in “e”, but it is shown in “c”. The same error is on the text (line 158)
Maybe a table describing all the 13 patients with the age, sex, localization of the tumors, and positivity for CK20, Syn, CrhA, NFP, Piezo1, and Piezo2 would better summarize the cases.
Author Response
“The authors describe the expression of Piezo channels in Merkel cell carcinoma. These markers were identified in Merkel cells, but this is the first study to address the question of whether MCC cells express these channels. This is an interesting study since novel biomarkers for diseases are potential targets for new drugs.”
- Thank you very much for your interest and support. Your suggestions are highly appreciated.
“Page 1, line 38: what is “pre/pro”? Would it be pre-B cells and pro-B cells?”
- Certainly, it was an error.
“Page 2, lines 67-79: “This could be of potential interest because the role of mechanosensitive ion channels in cancer, with diagnostic or prognostic purposes, but also a noticeable potential in therapy”. This sentence should be reformulated: “This could be of potential interest because of the role of mechanosensitive ion channels in diagnostic, prognostic, treatment in cancer””
- Thank you for the suggestion. It is clearer this way and we subsequently modified the sentence.
“Figure 2. The legend says that CK20 is demonstrated in “a” and “b”, but images show “b” as Piezo 2, and “e” as CK20. The same error is on the text (line 146)”
“Figure 3. The legend says that CK20 is demonstrated in “e”, but it is shown in “c”. The same error is on the text (line 158)”
- It´s true, we modified both figures to match the text.
“Maybe a table describing all the 13 patients with the age, sex, localization of the tumors, and positivity for CK20, Syn, CrhA, NFP, Piezo1, and Piezo2 would better summarize the cases.”
- We understand your suggestion, but considering that all the tumors have similar expression intensity with the employed antibodies and there are only two morphological patterns of expression, both cytoplasmic and often overlapping, we think this table provides little value.
Reviewer 3 Report
Dear Academic Publisher and dear authors, I have read with pleasure and attention your paper which I consider very valid. This is a study related to Merkel cell carcinoma (once called neuroendocrine carcinoma of the skin) on which cases the authors conducted immunohistochemical investigations for both the CK20, Synaptophysin, Chromogranin and Neurofilament Proteins markers, both for PIEZO1 and 2 The results are very interesting both from a prognosis and a research point of view.
Please be consistent: MMC or MCC
Author Response
“Dear Academic Publisher and dear authors, I have read with pleasure and attention your paper which I consider very valid. This is a study related to Merkel cell carcinoma (once called neuroendocrine carcinoma of the skin) on which cases the authors conducted immunohistochemical investigations for both the CK20, Synaptophysin, Chromogranin and Neurofilament Proteins markers, both for PIEZO1 and 2 The results are very interesting both from a prognosis and a research point of view.
Please be consistent: MMC or MCC”
- Thank you very much for your interest and support. The mentioned issue was corrected: MCC is the correct acronym for Merkel Cell Carcinoma.